# Efficient spin accumulation carried by slow relaxons in chiral tellurium

Evgenii Barts [1] ✉, Karma Tenzin [1,2] ✉ & Jagoda Sławińska [1] ✉

Efficient conversion between charge currents and spin signals is crucial for realizing magnet-free spintronic devices. However, the strong spin-orbit coupling that enhances this conversion also causes rapid spin dissipation, making spin signals difficult to control. Although modern materials science offers novel systems with diverse spin configurations of conduction electrons, understanding their fundamental limitations requires insights into the mechanisms behind the creation and relaxation of spin populations. In this study, we demonstrate that parallel spin-momentum entanglement at the Fermi surface of chiral tellurium crystals gives rise to slow collective relaxation modes, termed *relaxons*. These relaxons dominate the electrically generated spin and orbital angular momentum accumulation in tellurium, achieving an extraordinary 50% conversion efficiency, and are responsible for a long lifetime of the spin population. We show that the slow relaxons carrying spin density closely resemble the persistent helical spin states observed in GaAs semiconductor quantum wells. This similarity suggests that slow relaxons are a general phenomenon, potentially present in other chiral materials with strong spin-momentum locking, and could be used to generate and transmit spin signals with low heat losses in future electronics.

Spintronic devices that make use of both the charge and spin of electrons can provide high-performance and energy-efficient solutions for electronics[1]. However, progress in the field is hindered by the insufficient compatibility between semiconductors and ferromagnets used for spin injection and detection[2]. An alternative approach without ferromagnetic electrodes relies on the generation of spin signals via charge currents using charge-to-spin conversion mechanisms[3,4]. Nevertheless, the conversion efficiency in these processes typically does not exceed a few percent, which is far below the values required for practical devices and circuits[5]. Moreover, the strong spin-orbit coupling (SOC) that makes charge-to-spin conversion phenomena detectable has a detrimental effect on the generated spin signals. It creates an effective momentum-dependent magnetic field that acts on electrons traveling through the crystal, causing precession of the spin and its dephasing upon scattering[6]. Thus, using charge currents for efficient generation of spin signals that could survive over large distances remains a fundamental trade-off.

The idea of a peculiar spin precession mode called a persistent spin helix (PSH) was suggested as a remedy to protect spins from decoherence in the presence of strong SOC[7]. It relies on the engineering of spin-orbit interaction via external tuning of the system's symmetries. When inversion symmetry is lacking, SOC splits energy states, manifesting as a form of Zeeman interaction with a momentum-dependent magnetic field. Depending on symmetries, the SOC can be of Rashba or Dresselhaus type:

$$H_{\text{SOC}} = \alpha_{\text{R}}\left(k_x \sigma_y - k_y \sigma_x\right) + \alpha_{\text{D}}\left(k_x \sigma_x - k_y \sigma_y\right), \quad (1)$$

where $\sigma_{x,y}$ are Pauli matrices in spin space. If their coupling strengths are equal ($\alpha_{\text{R}} = \alpha_{\text{D}}$), the spin-orbit field becomes unidirectional and momentum-independent, yielding a configuration known as a persistent spin texture (PST)[8]. In real space, the spin of a moving electron is subjected to a controlled precession around this field, which results in a spatially-modulated periodic mode protected against decoherence.

[1]Zernike Institute for Advanced Materials, University of Groningen, 9747AG Groningen, The Netherlands. [2]Department of Physical Science, Sherubtse College, Royal University of Bhutan, 42007 Kanglung, Trashigang, Bhutan. ✉e-mail: e.barts@rug.nl; k.tenzin@rug.nl; jagoda.slawinska@rug.nl

Thus, PST allows for a spin-wave collective excitation characterized by an infinite lifetime[8,9], the hallmark of PSH, sketched in Fig. 1a. PSH with nanosecond lifetime was experimentally observed in GaAs quantum wells with balanced Rashba and Dresselhaus SOC[10], and its real space helical pattern was later imaged optically[11]. Despite its fundamental importance, PSH was not commonly explored for devices due to the need for fine-tuning of SOC parameters and temperature limitations of quantum wells.

Recently, the same mechanism for generating PST has been discovered in orthorhombic crystals, where the interplay of crystal symmetries naturally ensures equal strengths of Rashba and Dresselhaus interactions, enforcing a unidirectional spin polarization of states in momentum space[12]. Several materials with PST have been theoretically predicted[13–15] and suggested as ideal candidates for robust spin transport at room temperature. Surprisingly, PSH in solid-state materials has rarely been explored experimentally[16,17]. While semiclassical Boltzmann transport calculations suggest that systems with comparable Rashba and Dresselhaus couplings can support a finite electrically induced spin accumulation[18,19], linear response theory, supported by gauge symmetry arguments, predicts that this effect rapidly vanishes when these couplings are exactly balanced[20]. Given the lack of experimental evidence, efficiently generating spin accumulation by an electric current in materials with PST may be challenging, meaning it would still need to be injected using ferromagnets or excited optically. Such limitations raise important questions about the compatibility of PSH with all-electrical generation, manipulation, and detection of spin signals. Can we find materials with strong SOC enabling both the efficient electrical generation of spin accumulation and long-range spin transport via a mechanism similar to PSH?

In this Article, we show that bulk crystals of tellurium (Te) enable charge-to-spin conversion with unprecedented efficiency, reaching 50%. In contrast to systems with equal Rashba and Dresselhaus parameters, spin generation in Te is possible via the collinear Rashba-Edelstein effect, whereby an applied charge current induces accumulation of spins parallel to the current direction[21,22]. When the current flows along Te chains, we find that a PST aligned in the same direction boosts the spin accumulation efficiency and partially suppresses the back-scattering of electrons, as presented in the inset of Fig. 1b. This results in an extended spin lifetime, making Te suitable for both spin generation and transport in devices similar to the one shown in Fig. 1b. To quantify these effects, we use the concept of relaxons allowing for exact solutions of the Boltzmann transport equations. Our calculations of current-induced spin and orbital angular momentum accumulation quantitatively agree with nuclear magnetic resonance (NMR)

experimental data[23,24]. Furthermore, we reveal that the relaxation of the non-equilibrium electron population is characterized by two collective relaxation modes, but only the slower mode, resembling PSH in quantum wells, carries spin-momentum and governs the magnitude and lifetime of spin accumulation. Therefore, the generated spin signals relax at rates slower than the mean electron momentum relaxation rate, supporting long-range spin transport.

## Results and discussion
### Exact Boltzmann transport approach
To quantify the current-induced spin accumulation in tellurium crystals, we use the semiclassical Boltzmann equation[25]. It describes the time evolution of the electron distribution function, $f_{\boldsymbol{k}}$, under external stimuli:

$$\frac{\partial f_{\boldsymbol{k}}(t)}{\partial t} + (-e)\boldsymbol{E} \cdot \frac{\partial f_{\boldsymbol{k}}(t)}{\hbar \partial \boldsymbol{k}} = \left(\frac{\partial f_{\boldsymbol{k}}}{\partial t}\right)_{\text{col}}, \qquad (2)$$

where $\boldsymbol{k}$ is the reciprocal vector, $\boldsymbol{E}$ is the applied electric field, and $(-e) < 0$ is the electron charge. The collision integral on the right-hand side of Eq. (2) accounts for scattering processes with impurities. While the Boltzmann equation allows us to compute charge and spin transport properties, it is challenging to solve without approximations.

A common approach to solving this equation is the relaxation time approximation[25], given by $\left(\frac{\partial f_{\boldsymbol{k}}}{\partial t}\right)_{\text{col}} = -\delta f_{\boldsymbol{k}}/\tau$. Although this approximation often captures the key properties of transport phenomena, it has limitations[26]. Specifically, it predicts an exponential decay of the deviation from the equilibrium Fermi-Dirac distribution: $\delta f_{\boldsymbol{k}}(t) = f_{\boldsymbol{k}}(t) - f_{\boldsymbol{k}}^{(0)} \sim e^{-t/\tau}$, which can oversimplify the dynamics. Meanwhile, determining whether one or multiple characteristic timescales describe the dynamics of a system is more complex. This issue becomes more pronounced in materials where the spin relaxation time significantly exceeds the mean electron scattering time. Thus, the constant relaxation time approximation is particularly limited when describing materials with extended spin lifetimes.

To address this, we solve the Boltzmann equation exactly using a microscopic form of the collision integral[25]:

$$\left(\frac{\partial f_{\boldsymbol{k}}}{\partial t}\right)_{\text{col}} = -\sum_{\boldsymbol{k}'} W_{\boldsymbol{k}\boldsymbol{k}'}(f_{\boldsymbol{k}} - f_{\boldsymbol{k}'}), \qquad (3)$$

where the scattering probability, $W_{\boldsymbol{k}\boldsymbol{k}'}$, is given by Fermi's golden rule: $W_{\boldsymbol{k}\boldsymbol{k}'} = \frac{2\pi}{\hbar}|\langle \boldsymbol{k}'|H_{\text{int}}|\boldsymbol{k}\rangle|^2\delta(\varepsilon_{\boldsymbol{k}'} - \varepsilon_{\boldsymbol{k}})$. Here, $\varepsilon_{\boldsymbol{k}}$ is the band dispersion, and $H_{\text{int}}(\boldsymbol{r}) = U_{\text{imp}}\sum_a \delta(\boldsymbol{r} - \boldsymbol{r}_a)$ describes the interaction with static impurities,

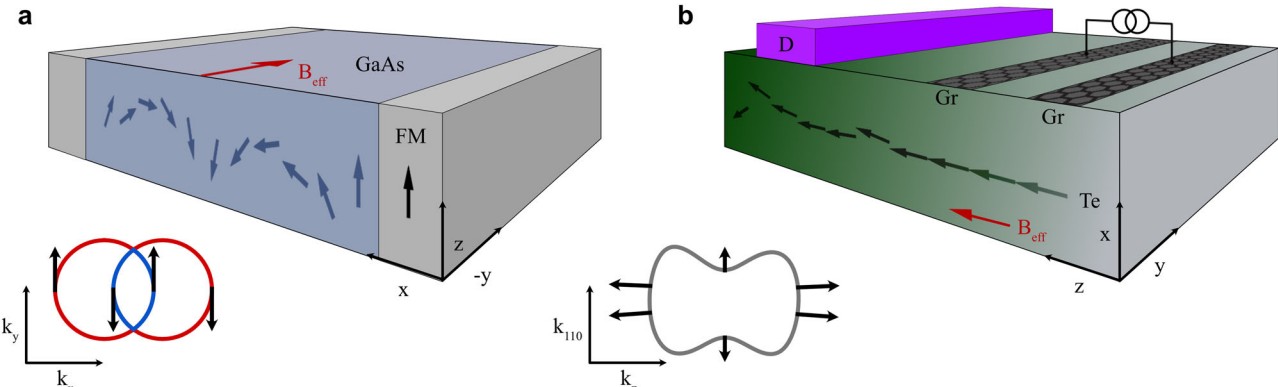

**Fig. 1 | Spin transport in GaAs quantum wells and in chiral Te crystals. a** Scheme of the coherent spin precession in GaAs with equal Rashba and Dresselhaus parameters ($\alpha_{\text{R}} = \alpha_{\text{D}}$). The inset illustrates the spin polarization of bands in this model, resulting in PST along $k_y$. The spins are injected from the ferromagnet and travel along the $x$ direction over a large distance. The effective spin-orbit magnetic field ($B_{\text{eff}}$) along the $y$ axis causes coherent spin precession in the $xz$ plane,

independently of scattering events. **b** Example of a device for generating, transmitting, and detecting spin accumulation in Te. The electric current is applied along the $z$ axis, and the induced spin accumulation along the current direction is transmitted toward the detector. Beyond the region with the electrodes, the spin accumulation slowly decays. The inset schematically illustrates the topmost valence band of Te, enabling the efficient spin transport.

modeled by delta function short-range potentials of strength $U_{\mathrm{imp}}$ and randomly distributed with a low density $n_{\mathrm{imp}}$. Further details of the model implementation can be found in Methods. The scattering amplitudes $\langle \boldsymbol{k}'|H_{\mathrm{int}}|\boldsymbol{k}\rangle$ depend on the spin, orbital, and sublattice degrees of freedom encoded within the states $|\boldsymbol{k}\rangle$, which enables accurate calculations of transport coefficients even in systems with strong entanglement of these degrees of freedom with momentum $\boldsymbol{k}$.

By introducing an ansatz $\delta f_{\boldsymbol{k}}(t) = e^{-t/\tau}\mathcal{V}_{\boldsymbol{k}}$, we transform the Boltzmann equation at $\boldsymbol{E} = 0$ into an eigenvalue problem:

$$\sum_{\boldsymbol{k}'} \mathcal{K}_{\boldsymbol{k}\boldsymbol{k}'}\mathcal{V}_{\boldsymbol{k}'} = \Gamma \mathcal{V}_{\boldsymbol{k}}, \tag{4}$$

where $\mathcal{K}_{\boldsymbol{k}\boldsymbol{k}'}$ is the relaxation matrix, defined as

$$\mathcal{K}_{\boldsymbol{k}\boldsymbol{k}'}/\tau_0 = -W_{\boldsymbol{k}\boldsymbol{k}'} + \delta_{\boldsymbol{k}\boldsymbol{k}'}\sum_{\boldsymbol{k}''} W_{\boldsymbol{k}\boldsymbol{k}''}. \tag{5}$$

We introduce dimensionless time units through a characteristic scattering time, $\tau_0 = \hbar/(\pi n_{\mathrm{imp}}U_{\mathrm{imp}}^2\rho_0)$. For systems with low SOC, $\tau_0$ would correspond to the mean scattering time. Here, $\rho_0$ is the electron density of states at the chemical potential equal to energy $\varepsilon_0$. We adopt $\varepsilon_0 = -20$ meV, relative to the valence band top, aligning with natural hole doping in Te ($7.4 \times 10^{17}$/cm$^3$ [22]).

The eigenvector of the relaxation matrix, $\mathcal{V}_{\boldsymbol{k}}$, and its eigenvalue, $\Gamma = \tau_0/\tau$, define a 'relaxon'–a collective relaxation mode with the dimensionless relaxation rate $\Gamma$[27,28]. Relaxons have been mostly used to quantify thermal transport in crystals, where they are collective phonon excitations or wavepackets acting as elementary heat conductors[27,29,30], and recently, to calculate spin relaxation time and diffusion length for free electron gas with anisotropic Weyl-type SOC[28]. In our framework, each relaxon represents a wavepacket of particle-hole excitations above the Fermi sea with a well-defined transport lifetime.

The relaxons form a complete orthonormal basis set, evidenced by the completeness relation $\sum_{\boldsymbol{k}} \mathcal{V}_{\boldsymbol{k}}^i \mathcal{V}_{\boldsymbol{k}}^j = \delta_{ij}$, which makes it convenient for finding exact solutions of the Boltzmann equation. Namely, any non-equilibrium electron distribution can be expressed as a linear combination of relaxons[29]:

$$\delta f_{\boldsymbol{k}}(t) = \sum_i A_i(t)\,\mathcal{V}_{\boldsymbol{k}}^i, \tag{6}$$

where $i$ labels an individual relaxon, and its amplitude $A_i(t)$ relaxes with the rate $\Gamma_i$ as $A_i(t) = A_i(t=0)e^{-\Gamma_i t/\tau_0}$. Such a spectral decomposition allows us to calculate transport coefficients within the linear regime. For instance, inserting the decomposed $\delta f_{\boldsymbol{k}}$ into the steady-state Boltzmann equation yields the spectral amplitudes of an electrically induced electron population:

$$A_i(0) = \tau_i \sum_{\boldsymbol{k}} e\boldsymbol{E} \cdot \boldsymbol{v}_{\boldsymbol{k}}\left(-\frac{\partial f_{\boldsymbol{k}}^{(0)}}{\partial \varepsilon_{\boldsymbol{k}}}\right)\mathcal{V}_{\boldsymbol{k}}^i, \tag{7}$$

where $\boldsymbol{v}_{\boldsymbol{k}} = \frac{\partial \varepsilon_{\boldsymbol{k}}}{\hbar \partial \boldsymbol{k}}$ is the velocity. For this derivation, we used the completeness relation after multiplying the Boltzmann equation by $\mathcal{V}_{\boldsymbol{k}}^j$ and summing over $\boldsymbol{k}$. Furthermore, adopting dimensionless time units effectively removes an irrelevant overall factor in Eq. (7), which cancels out the term $n_{\mathrm{imp}}U_{\mathrm{imp}}^2$. Consequently, the dimensionless relaxation time and response coefficients depend solely on the intrinsic properties of the electronic states, remaining independent of any disorder characteristics.

Finally, the current-induced spin and orbital angular momentum accumulation can be generally expressed as:

$$M_a = \chi_{ab}j_b, \tag{8}$$

where $a$, $b = x$, $y$, $z$ denote the components of the induced magnetization $M_a$ and electron current density $j_b$. The Rashba-Edelstein response tensor is given by ref. 31:

$$\chi_{ab} = \frac{\sum_{\boldsymbol{k}}\langle \hat{\mu}_a\rangle_{\boldsymbol{k}}\delta f_{\boldsymbol{k}}}{-e\sum_{\boldsymbol{k}}v_{\boldsymbol{k}}^b\delta f_{\boldsymbol{k}}}, \tag{9}$$

where $\langle \hat{\mu}_a\rangle_{\boldsymbol{k}}$ is the magnetic dipole moment expectation value for the state $\boldsymbol{k}$, with $\hat{\mu}_a = (2\hat{S}_a + \hat{L}_a)\mu_B$. Here, $\hat{S}_a = \sigma_a/2$ is the spin-1/2 operator, and $\hat{L}_a = -i\varepsilon_{abc}|p_b\rangle\langle p_c|$ captures the on-site orbital angular momentum contribution from the atomic $p$ orbitals. The denominator corresponds to the generated charge current. Based on Eqs. (6)–(9), we calculate the current-induced accumulation and its lifetime in bulk Te. We note that our approach is particularly suitable for calculating measured quantities because it contains no free parameters. The parameter $\tau_0$ cancels out in the Rashba-Edelstein tensor, as it contributes linearly to both the numerator and denominator. Overall, our exact Boltzmann framework offers a robust methodology for accurately calculating spin transport coefficients in arbitrary materials.

## Efficiency of the current-induced spin and orbital angular momentum accumulation in chiral Te crystals

Right- and left-handed Te crystals belong to the space symmetry groups $P3_121$ and $P3_221$, respectively. The crystal symmetry enforces current-induced magnetization parallel to charge current, thus allowing only non-zero components $\chi_{zz}$ and $\chi_{xx} = \chi_{yy}$[32]. Because the two enantiomers are connected through inversion, their Rashba-Edelstein response tensors only differ by sign, and we restrict our analysis to the right-handed Te. The details of its structure and calculated electronic properties are shown in Fig. 2a–d. The spin- and orbital-resolved electronic states are obtained from density functional theory (DFT) and tight-binding calculations, as described in Methods.

Figure 2e shows the magnetization $M_z$ induced by an electric current along the $z$ axis ($j_z = 82$ A cm$^{-2}$) as a function of the chemical potential, demonstrating good agreement with the NMR experimental data, which indicated $M_z = 1.3 \times 10^{-8}\mu_B$ per Te for this value of current[23]. We highlight a large improvement compared to the previous theoretical calculations, which underestimated the magnetization by an order of magnitude[21–23]. While one could presume a role of the orbital contribution, Fig. 2g shows that the orbital contribution alone only doubles the total magnetization for the wide range of chemical potentials that could result from the realistic hole doping levels. The magnetization $M_z$ already incorporates the $g$-factors (2 for spin, 1 for orbital momentum), making the spin and orbital contributions nearly equal throughout the considered chemical potential range. The accuracy primarily comes from incorporating the $k$-state dependent relaxation time, achieved via the exact solution of the Boltzmann equation. Note that the observed magnetization increase near the valence band top is not fully reliable due to the division of two small numbers at very low hole populations in the magnetization defined in Eqs. (8), (9). Nevertheless, the spin accumulation exhibits the expected behavior at these small values of the chemical potential, as shown in Fig. 2g. Furthermore, an electric current along the $x$ axis ($j_x = 82$ A cm$^{-2}$) induces magnetization $M_x$ smaller than $M_z$ by an order of magnitude, showing strong anisotropy relative to the screw axis. The temperature was set to $T = 10$ K for our calculations, but the results are similar at higher temperatures.

The remaining discrepancy between the experimental and theoretical values of $M_z$ remains an important open question. In our analysis, we focus on how the accurate treatment of the relaxation times of individual carriers influences spin accumulation and its lifetime. Specifically, we calculate only the on-site orbital contribution to magnetization, commonly referred to in the literature as the atomic-centered approximation, while omitting intersite (or itinerant) contributions. To achieve more accurate predictions of magnetization, it is necessary to

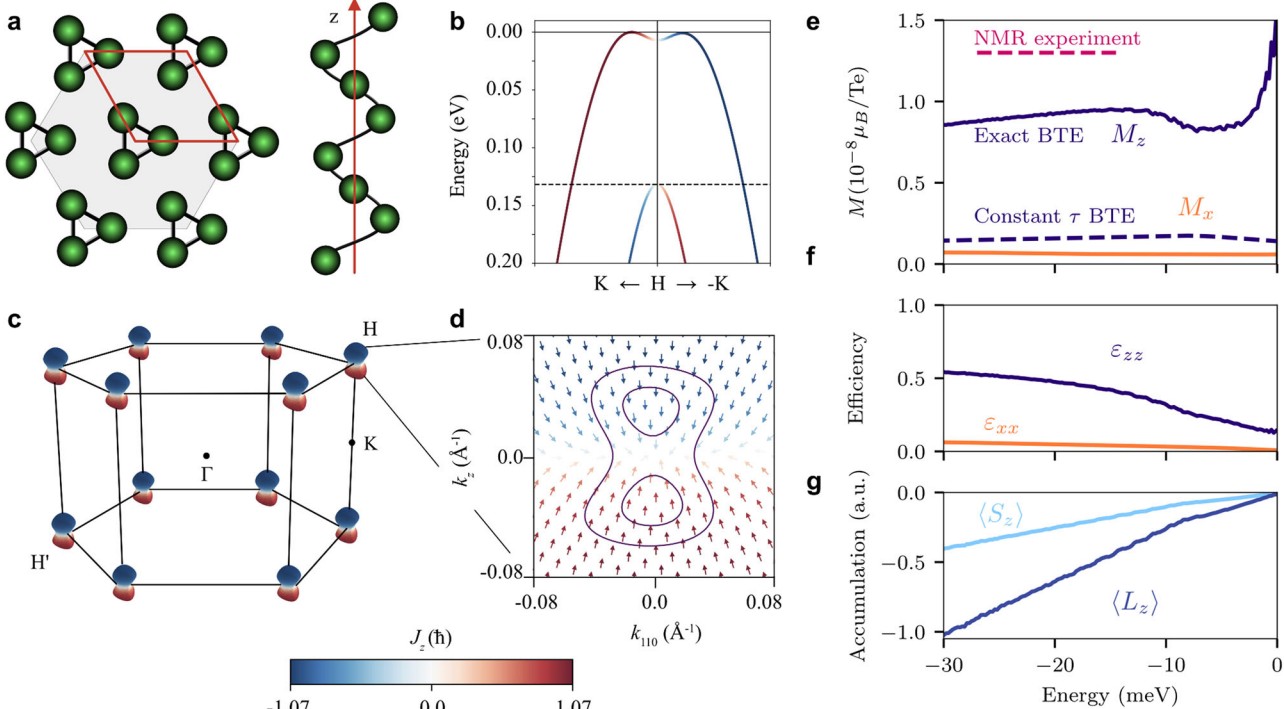

**Fig. 2 | Crystal structure, electronic properties, and charge-to-spin conversion in tellurium. a** Crystal structure of elemental Te. The helicity of an isolated chain along the *z* direction corresponds to right-handed Te. **b** Calculated valence band structure along the $k_z$ axis close to the *H* point in the Brillouin zone. The color encodes the *z* component of the total angular momentum. Its *x* and *y* components are much smaller in magnitude and are not shown. **c** The three-dimensional Fermi surface at $E = -20$ meV consists of six dumbbell-shaped hole pockets centered in the corners of the hexagonal Brillouin zone. **d** A single Fermi pocket at the *H* point projected onto a plane parallel to the *z* axis, where the arrows represent the total angular momentum directions. The constant energy contours at $E = -5$ meV (inner)

and $E = -20$ meV (outer) are shown. **e** Current-induced magnetization $M_z$ and $M_x$ per Te atom induced by an electric current along the *z* and the *x* axis, respectively, ($j_{x,z} = 82$ A cm$^{-2}$) calculated for different chemical potential values. In the realistic doping region, the red dashed line indicates the induced magnetization estimated from NMR[23]. The solid lines represent our calculated values using exact solutions of the Boltzmann transport equations (BTE), while the dashed dark line shows the $M_z$ value based on the relaxation time approximation[21]. **f** Charge-to-spin conversion efficiency vs chemical potential. **g** Spin and orbital contributions to angular momentum density without including g-factors.

include additional factors, such as wavepacket self-rotation and corrections to the density of states in an applied magnetic field–both accounted for by the modern theory of orbital magnetization[33–36]. They are relevant for 5*p* orbitals, given their strongly delocalized nature in Te[37], where previous studies suggested sizable contributions from wavepacket self-rotation[38,39]. Other extrinsic contributions to magnetization, such as from side-jump and skew scattering effects, become relevant only at higher hole densities[38]. These effects will be explored in more detail in future studies.

Although the induced magnetization is convenient for comparison with the NMR measurements, for nanodevices that rely on spin transport, the charge-to-spin conversion efficiency is a more important figure of merit. We define efficiency as:

$$\varepsilon_{zz} = \frac{\sum_{\boldsymbol{k}} \langle \hat{J}_z \rangle_{\boldsymbol{k}} \delta f_{\boldsymbol{k}}}{\sum_{\boldsymbol{k}} |\delta f_{\boldsymbol{k}}|}, \qquad (10)$$

where $\hat{J}_z = (\hat{S}_z + \hat{L}_z)/J$ is the *z* component of the total angular momentum polarization operator, normalized with $J = 3/2$, and the electric field is applied along the *z* axis. Importantly, $\varepsilon_{zz}$ has a transparent physical interpretation, resembling an intuitive efficiency definition $(N_\uparrow - N_\downarrow)/(N_\uparrow + N_\downarrow)$, where $N_\sigma$ is the non-equilibrium deviation of the total number of electrons with spin polarization $\sigma$. Whereas the numerator in Eq. (10) represents the total angular momentum polarization, its denominator quantifies the magnitude of the current-induced shift in the distribution function. In the quasi-classical picture, $\sum_{\boldsymbol{k}} |\delta f_{\boldsymbol{k}}|$ accounts for the induced imbalance between forward- and back-moving electron populations in momentum space. The efficiency is normalized from 0 to 1, where the

former indicates no spin signal, and the latter describes the ideal situation of maximum accumulation with opposite spins of forward- and back-moving electrons.

Figure 2f shows the charge-to-spin conversion efficiency of Te calculated as a function of the chemical potential. Tellurium demonstrates exceptional efficiency, reaching 50% at realistic hole doping levels ($7.4 \times 10^{17}$/cm$^3$) that correspond to the chemical potential of ~ $-20$ meV[22]. This value is much higher than in most known materials with strong SOC[40]. We note that lower efficiencies of 20–40% have been recently reported for Te in the context of possible chirality-induced spin selectivity based on spin current calculations in a ballistic transport regime[41,42]. However, our approach is more suitable for sample sizes larger than the mean free electron path, as it considers nonballistic diffusive transport by including disorder.

## Spin accumulation lifetime and relaxon spectra

We now examine the time dependence of spin accumulation by analyzing the relaxon spectrum. Figure 3 presents the spectral decomposition of the current-induced shift in the distribution function. When an electric current is applied along the *z* axis (see Fig. 3a), the spectrum has two pronounced peaks corresponding to collective relaxation modes: a fast mode with the relaxation rate $\Gamma \approx 0.5$ and a slow mode with $\Gamma \approx 0.3$ followed by a tiny satellite peak with $\Gamma \approx 0.22$, as shown in Fig. 3c. The contributions to any observable from different collective relaxation modes can be independently calculated using the relaxon spectral decomposition in Eq. (6). For example, the average spin density is given by $S_z(t) = \sum_{\boldsymbol{k}} \langle \hat{S}_z \rangle_{\boldsymbol{k}} \delta f_{\boldsymbol{k}}(t)/V = \sum_i \langle \hat{S}_z \rangle_i A_i(t)$, where $\langle \hat{S}_z \rangle_i = \sum_{\boldsymbol{k}} \langle \hat{S}_z \rangle_{\boldsymbol{k}} \mathcal{V}^i_{\boldsymbol{k}}/V$ is the spin polarization of the *i*-th relaxon.

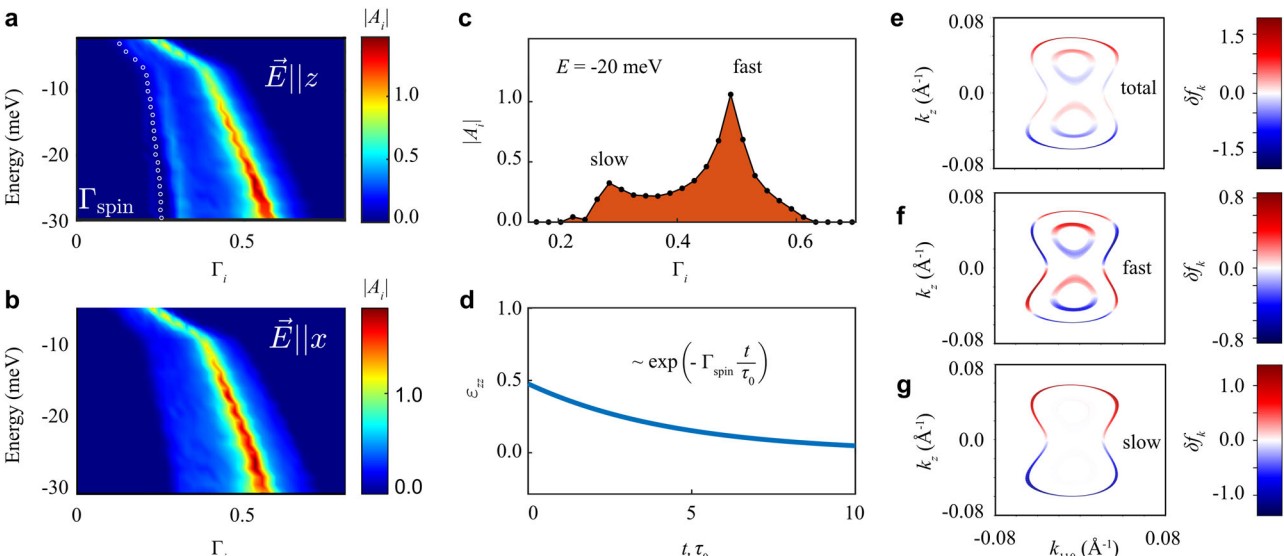

**Fig. 3 | Current-induced spin accumulation in Te in the relaxon basis.**
**a, b** Spectral decomposition at different values of chemical potential when an electric field $\vec{E}$ is applied along the $z$ and the $x$ axis, respectively. The spectral amplitudes are color-coded, and white dots show the calculated spin relaxation rate. For $\vec{E} \parallel z$ and energy $E = -20$ meV, **c** shows the spectrum, and **d** the spin

density time dependence, normalized as charge-to-spin conversion efficiency.
**e** The non-equilibrium deviation of the total distribution function $\delta f_k$ calculated near energies $-5$ meV (inner contour) and $-20$ meV (outer contour).
**f, g** Contributions to $\delta f_k$ from relaxons with $\Gamma_i > 0.33$ and $\Gamma_i < 0.33$, respectively.

We find that the main contribution to $M_z$ comes from the relaxons belonging to the slow modes (that is, $\Gamma_i < 0.33$). This is further confirmed by the spin relaxation time, calculated by fitting the spin time dependence to a single exponential decay model (see Fig. 3d). The resulting spin relaxation rate, marked by white dots in Fig. 3a, closely follows the slow relaxation modes, demonstrating their dominant role in governing spin accumulation. Conversely, when the electric current flows along the $x$ axis (see Fig. 3b), the slow modes vanish. The low calculated current-induced magnetization $M_x$ suggests that the fast mode corresponds to the mean electron scattering time, *i.e.*, a normal relaxation mode.

To clarify the origin of the slow mode, we visualize $\delta f_k$ on the Fermi pockets near the $H$ point. Figure 3e displays the $-20$ meV (and $-5$ meV) Fermi contours, where the color quantifies $\delta f_k$. Figure 3f, g further isolate the fast and slow modes by calculating the contributions to $\delta f_k$ from relaxons with $\Gamma_i > 0.33$ and $\Gamma_i < 0.33$, respectively. For $E = -20$ meV, the distribution $\delta f_k$ associated with the slow mode, where the spin polarization of induced holes is opposite to that of electrons, is identical to the angular momentum polarization pattern of electron states in Te (see Fig. 2d). This alignment of the slow mode distribution $\delta f_k$ with the spin polarization texture in $\boldsymbol{k}$-space reduces backscattering between the tips of the dumbbell-shaped pocket, thereby slowing this mode's relaxation. Since nonmagnetic impurity scattering does not flip spin, the scattering probability $W_{\boldsymbol{k}\boldsymbol{k}'} \sim |\langle \boldsymbol{k}'|\boldsymbol{k}\rangle|^2$ is diminishing when the states $|\boldsymbol{k}\rangle$ and $|\boldsymbol{k}'\rangle$ have opposite spin polarizations. Nonetheless, the suppression is only partial because the spin polarization of the electron states is partial. The total angular momentum of these states never attains its maximum value of 3/2, as illustrated in Fig. 2d, in agreement with the angle-resolved photoemission spectroscopy (ARPES) results[43]. The contour at $E = -5$ meV shows a different scenario, where the slow mode is nearly absent in the spectral decomposition (see Fig. 3g), resulting in much smaller current-induced accumulation (Fig. 2g) and shorter spin lifetime (Fig. 3a).

The spin relaxation mechanism in Te is particularly interesting. Notably, the conventional Dyakonov-Perel mechanism[44] is negligible in this context. Interband transitions, which could otherwise contribute to spin relaxation within coherent dynamics, are suppressed due to the large band splitting ($\approx 100$ meV) compared to other energy scales in the

system. As a result, coherent spin relaxation processes can, to a first approximation, be ignored in Te. As spin relaxation in Te arises from collisions with nonmagnetic impurities, it resembles the Elliot-Yafet mechanism[45,46] and can be thought of as a generalization of this mechanism for materials with strong SOC. In chiral Te, the strong SOC introduces significant mixing of opposite spin states in Bloch electrons, which would typically reduce spin lifetime. Yet, this suppression is counterbalanced by the strong spin and orbital polarization of carriers near the H point (see Fig. 2a–d and Supplementary Fig. 1). The structural chirality of Te imprints this polarization onto wavefunctions, aligning electrons velocities with their spin polarizations. This spin protection mechanism, stemming from the suppressed backscattering of spin-polarized states, bears the closest resemblance to the behavior of protected edge states in graphene[47] and topological insulators[48]. This analogy makes it promising to study quantum-geometrical aspects[49,50] of spin accumulation and its lifetime in chiral materials such as Te[51–53], especially in light of recently observed in ARPES experiments in-gap states[54] that are similar to the robust bound states localized at the chain boundaries in the Su-Schrieffer-Heeger model[55].

These observations suggest that the generated spin accumulation can propagate over long distances. The spin diffusion length is approximately the product of the spin relaxation time and the Fermi velocity. In Te, it is estimated to be about three times the mean free path (see Fig. 3a). Recent magnetoresistance experiments on Te nanoflakes reported mean free paths of 22–34 nm[56], corresponding to spin diffusion lengths of 66–102 nm. This range is comparable to the phase coherence and the spin-orbit relaxation lengths, which reach up to 100 nm based on weak localization model calculations. Such spin diffusion length scales are promising for crafting efficient spintronic devices that make use of non-local spin transport in two-dimensional heterostructures based on Te thin films or tellurene. A potential all-electrical spintronic device could operate in a non-local spin diffusive regime: in the 'injector' region, an electric current along the chain induces parallel magnetization, and in the 'conductor' region, spin accumulation diffuses over hundreds of nanometers towards the 'detector' region, where the spin signal is measured through spin-to-charge conversion (see Fig. 1b).

However, a direct mapping between relaxon lifetimes and experimental transport coefficients remains an open question for future research. Based on the mean free path (≈30 nm) and the Fermi velocity (≈ 30 nm/ps), we estimate the mean scattering time to be -1 ps, a typical value for semiconductors. This suggests that the spin relaxation time in Te would be -3 ps, which is significantly shorter than the typical Dyakonov-Perel and Elliot-Yafet timescales in *III-V* semiconductors[57,58], ranging from 10 to 100 ps at similar high carrier densities ($10^{16}$–$10^{18}$ cm$^{-3}$), and than that extending up to nanoseconds in GaAs quantum wells that host PSH[10].

Finally, we highlight a noticeable correlation between slow relaxons and charge-to-spin conversion efficiency. Long-living relaxons contribute more to the distribution function and, consequently, to magnetization, as indicated by the linear $\tau$-dependence of relaxon amplitudes in Eq. (7). The slow relaxons, with lifetimes around $3\tau_0$, carry spin and angular momentum, leading to a threefold enhancement in magnetization compared to the constant relaxation time approximation. However, Eq. (7) also includes the relaxon eigenvector, which introduces a correction to our estimate due to the complex spectrum of relaxons. This spectrum contains two peaks representing collective relaxation modes, which resemble relaxon wavepackets. Unlike previous studies, our reported order-of-magnitude magnetization enhancement comes from accounting for the exact amplitudes, shapes, and widths of these wavepackets, as captured by the exact mapping of the initial distribution function onto the relaxon basis. Remarkably, the charge-to-spin conversion efficiency is amplified by the high spectral density of spin-carrying slow relaxons. This efficiency is mainly determined by the ratio of spectral weights between the slow and fast wavepackets. Thus, to maximize the efficiency, it is crucial to identify materials that host slower relaxons with higher spectral amplitudes, a key avenue for future research.

## Low-energy model and persistent spin helix
In the analysis above, we solved the Boltzmann transport equations exactly using ab initio wavefunctions that describe low-energy hole states in Te crystals, which revealed efficient spin accumulation with extended lifetime. However, the distinctive slow relaxons can be a general phenomenon in a broader class of chiral materials. To explore this further, we examine the relationship between the spin texture of the valence band and the slow relaxation mode using a $k \cdot p$ model that describes the low-energy behavior of holes in Te[59–61]. We show that this slow mode corresponds to the persistent spin helix, similar to those observed in GaAs quantum wells[10].

The two upper-lying valence bands of Te are described by the following two-state Hamiltonian:

$$\hat{\mathcal{H}}_k = - A k_z^2 - B\left(k_x^2 + k_y^2\right) + \beta k_z \hat{\tau}_z + \Delta \hat{\tau}_x, \qquad (11)$$

where $\hat{\tau}_z$ and $\hat{\tau}_x$ are Pauli matrices in pseudospin space. This model is minimal for describing the opening of the energy gap between the two valence bands, as shown in Fig. 2b. The Hamiltonian becomes diagonal in the basis of states $|\psi_\pm\rangle$, where $|\psi_+\rangle$ corresponds to the upper band and $|\psi_-\rangle$ the lower band. While detailed model parameters, band dispersion, and wavefunctions are discussed in Methods, it is crucial to note that the pseudospin polarization, as given by $\langle\psi_+|\hat{\tau}_z|\psi_+\rangle \sim \beta k_z$, largely reflects the spin polarization, making the pseudospin quantum number useful for interpreting the slow relaxation mode in connection with the spin texture of the electron states. Next, we will solve Boltzmann transport equations within the $k \cdot p$ model to generalize the concept of slow relaxons.

The relaxon spectral decomposition of the distribution function in the Rashba-Edelstein effect shows a strong dependence on $\Delta$. Figure 4 shows the impact of varying the magnitude of $\Delta$ while keeping a constant chemical potential of $E = -20$ meV. By analogy with the previous section, we calculate the average pseudospin value,

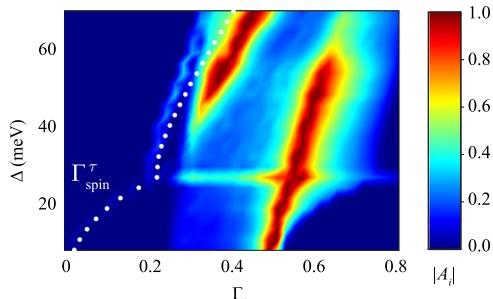

**Fig. 4 | Current-induced pseudospin accumulation in Te calculated using the low-energy model.** Spectral decomposition at different values of $\Delta$, which represents the energy gap between the two valence bands. The electric field is applied along the $z$ axis. The spectral amplitudes are color-coded, and white dots show the calculated pseudospin relaxation rate. The spectrum at each $\Delta$ is normalized to the unit at its maximum value. The Fermi level is set at $E_F = -20$ meV.

$\sum_k \langle\hat{\tau}_z\rangle_k \delta f_k(t)/V$, and its relaxation time. For realistic values describing Te bands, with $\Delta \approx 60$ meV, the pseudospin relaxation rate and the two relaxation peaks are in agreement with those shown in Fig. 3a. As $\Delta$ decreases, the spectral weight of the slow mode diminishes and nearly disappears at $\Delta \approx 25$ meV. This threshold matches an abrupt change in the Fermi contours topology from a dumbbell shape (see Fig. 2d) to two separate circles. When the neck of the dumbbell shrinks to zero, the fast relaxation mode dominates.

The pseudospin relaxation rate, indicated by white dots in Fig. 4, closely follows a third tiny peak in the spectrum. This third peak, a satellite of the slower mode, is visible in the $\Delta$-region between 30 and 60 meV. Although the slow mode disappears through the crossover region at $\Delta \approx 25$ meV, the pseudospin polarization does not completely disappear but gradually reduces as $\Delta$ decreases. The Rashba-Edelstein response diminishes and ultimately disappears in the small $\Delta$ limit, as predicted for uniaxial spin textures in GaAs[20]. Naturally, we confirm that the pseudospin relaxation rate eventually drops to zero as $\Delta$ approaches small values.

These observations indicate that the slow mode depicted in Fig. 4 tracks the $\Delta$-evolution of the persistent spin helix state. If $\Delta$ were exactly zero, our model would be equivalent—up to a rotation of coordinates—to the system with equal Rashba and Dresselhaus parameters, where the PSH has been observed. There is, however, one important difference: in GaAs, the spins and the shift in momentum space are orthogonal to each other, whereas in Te, they are parallel, aligning the magnetization induced by the applied current. Despite this difference, the argument suggesting emergent SU(2) spin rotational symmetry[8] still applies. It suggests that the PSH state manifests as a spiral spin state, where the spins lie in the plane that is orthogonal to the PSH wavevector. Conversely, a finite $\Delta$ breaks the SU(2) symmetry, introducing non-collinearity in the momentum space spin texture and thereby imposing a finite lifetime of the spin helix. Thus, we conclude that the slow mode corresponds to the PSH, whose finite lifetime is due to the non-zero gap $\Delta$.

In summary, we demonstrate that the highly efficient charge-to-spin conversion in chiral tellurium is governed by slow collective relaxation modes, which are crucial for both the electrical generation of spin signals with a record efficiency of 50% and their propagation over long distances up to 100 nm—two effects previously thought to be fundamentally incompatible. We explain the physical origin of the slow relaxons by the parallel spin-momentum entanglement at the Fermi surface, which reduces the back-scattering of charge carriers, and we suggest that they are also present in other chiral materials with strong spin-momentum locking. This concept opens up new avenues for spintronic devices based on chiral crystals, enabling efficient generation and transmission of spin signals over long distances.

## Methods

### First-principles calculations for Te

We performed DFT calculations for bulk Te using the Quantum Espresso package[62,63]. We employed the Perdew, Burke, and Ernzerhof generalized gradient approximation for exchange-correlation functional as well as fully relativistic pseudopotentials[64–66]. The electron wavefunctions were expanded in a plane-wave basis with the energy cutoff of 80 Ry. The structure with a hexagonal unit cell containing three atoms and lattice constants $a = 4.52$ Å and $c = 5.81$ Å was adopted after the full optimization with the energy and force convergence criteria set to $10^{-5}$ Ry and $10^{-4}$ Ry/bohr, respectively. The BZ was sampled following the Monkhorst-Pack scheme with the $k$-grids of $22 \times 22 \times 16$ and a Gaussian smearing of 0.001 Ry. The electronic structure was additionally corrected with the Hubbard parameter ($U_{5p} = 3.81$ eV) calculated self-consistently via the ACBN0 approach[67]. The spin-orbit interaction was included self-consistently in the calculation. As a post-processing step, we used the open-source Python code PAOFLOW[68,69] to project DFT wavefunctions onto the pseudoatomic orbitals and construct tight-binding Hamiltonians[70–72]. We further interpolated these Hamiltonians to denser $k$-grids of $101 \times 101 \times 101$ around the $H$ high-symmetry point of the BZ ($k_z \in k_z^H \pm 0.06 \cdot 2\pi/c$ and $k_{x,y} \in k_{x,y}^H \pm 0.03 \cdot 2\pi/a$) to accurately calculate the response coefficients. Transport calculations are performed using only $p$ orbitals since they fully describe the valence band states close to the H point.

### Boltzmann transport equations

The ab initio wavefunctions and energy bands in the pseudoatomic basis are used to construct the relaxation matrix $\mathcal{K}_{\boldsymbol{kk'}}$, defined in Eq. (5). The relaxation matrix incorporates the scattering probability, $W_{\boldsymbol{kk'}}$, which encodes the spin, orbital, and sublattice degrees of freedom within the scattering amplitudes. These amplitudes are given by $\langle \boldsymbol{k'}|H_{\text{int}}|\boldsymbol{k}\rangle = \frac{U_{\text{imp}}}{V}\sum_a e^{-i(\boldsymbol{k'}-\boldsymbol{k})\cdot\boldsymbol{r}_a}\langle \boldsymbol{k'}|\boldsymbol{k}\rangle$, where $V$ is the total volume. To obtain the scattering probability using Fermi's golden rule, we calculate the square of the matrix elements as: $|\langle \boldsymbol{k'}|H_{\text{int}}|\boldsymbol{k}\rangle|^2 = \frac{U_{\text{imp}}^2 n_{\text{imp}}}{V}|\langle \boldsymbol{k'}|\boldsymbol{k}\rangle|^2$. We regularize the Dirac delta function using the Lorentzian form: $\delta(\varepsilon_{\boldsymbol{k'}} - \varepsilon_{\boldsymbol{k}}) = \frac{1}{\pi}\frac{\Delta_E}{\Delta_E^2 + (\varepsilon_{\boldsymbol{k'}}-\varepsilon_{\boldsymbol{k}})^2}$, where the width $\Delta_E$ is set to 0.2 meV in our calculations. This regularization is also used to calculate the density of states: $\rho_0 = \frac{1}{V}\sum_{\boldsymbol{k}}\delta(\varepsilon_0 - \varepsilon_{\boldsymbol{k}})$. To diagonalize the relaxation matrix for each calculation at a given Fermi level, we include electron states in a small window near the Fermi level ($\pm 2$ meV). By numerically diagonalizing the relaxation matrix, we obtain a set of eigenvalues, $\Gamma_i$, and eigenstates, $\mathcal{V}_{\boldsymbol{k}}^i$, which form the relaxon basis. Using this basis, we calculate the spectral amplitudes $A_i(0)$, as defined in Eq. (6), to determine the variation in the distribution function $\delta f_{\boldsymbol{k}}$ due to an electric current.

### Low-energy Hamiltonian near the H point

Diagonalizing the Hamiltonian in Eq. (11) gives the upper and lower valence band dispersions:

$$E_{\pm} = -Ak_z^2 - B\left(k_x^2 + k_y^2\right) \pm \sqrt{\Delta^2 + \beta^2 k_z^2}, \qquad (12)$$

described by the wavefunctions:

$$|\psi_{\pm}\rangle = \left(\sqrt{1+\chi}|1/2\rangle \pm \sqrt{1-\chi}|-1/2\rangle\right)/\sqrt{2} \qquad (13)$$

with $\chi(k_z) = \beta k_z/\sqrt{\Delta^2 + \beta^2 k_z^2}$ and $\beta > 0$ (or $\beta < 0$) corresponding to the left-handed (or right-handed) Te. The pseudospin polarization of the upper band, given by $\langle \psi_+|\hat{\tau}_z|\psi_+\rangle = \chi$, closely follows the spin polarization shown in Fig. 2d. Note that the pseudospin is often referred to as $|\pm 3/2\rangle$ states in the literature, convenient for multiband

representations described by higher effective spins. This different labeling does not affect our calculations. The resulting upper and lower band wavefunctions, $|\psi_+\rangle$ and $|\psi_-\rangle$, transform as irreducible representations of the symmetry group of the $H$ point: $H_4$ and $H_5$, respectively. For the transport calculations within the $k \cdot p$ model, we use the following parameters: $A = 32.6$ eV Å$^2$, $B = 36.4$ eV Å$^2$, $\beta = 2.47$ eV Å, and $\Delta = 63$ meV[23,61]. In these calculations, the disorder matrix elements are diagonal in the pseudospin space, even though in reality, they are diagonal in the spin space. Nevertheless, we expect the results to qualitatively agree with those obtained from the ab initio wavefunctions, given that the pseudospin polarization closely follows the spin and orbital angular momentum polarization of Te bands.

## Data availability

The data associated with the manuscript are available via DataverseNL, https://doi.org/10.34894/MYORSD.

## Code availability

The MATLAB code used in this study to obtain exact solutions of the Boltzmann equation is available at GitHub (https://github.com/EBarts/Boltzmann_Solver).

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

## Acknowledgements

We thank B.J. van Wees, M. Mostovoy, M.A. Loi, K. Sundararajan, A. Roy, S. Tirion, S.B. Kilic, F. Cerasoli, and B. van Dijk for the helpful discussions. J.S. acknowledges the Rosalind Franklin Fellowship from the University of Groningen. E.B. and J.S. acknowledge the grant of the Dutch Research Agenda (NWA) under the contract NWA.1418.22.014 financed by the Dutch Research Council (NWO). The calculations were carried out on the Dutch national e-infrastructure with the support of SURF Cooperative (EINF-5312) and on the Hábrók high-performance computing cluster of the University of Groningen.

## Author contributions

E.B. performed the theoretical analysis, including the conceptualization of the Boltzmann solver and the studies of the persistent spin helix using the low-energy model. K.T. performed the computational part of the research in the framework of the density functional theory and PAO Hamiltonians. J.S. conceived and supervised the project. All the authors contributed to the data analysis and discussions.

## Competing interests

The authors declare no competing interests.
