## [Peer Review File · Nature Communications]

Efficient spin accumulation carried by slow relaxons in chiral tellurium

Corresponding Author: Professor Jagoda Sławińska

Version 1:

Reviewer comments:

Reviewer #1

(Remarks to the Author)

In this work, based on the advanced Boltzmann transport theory beyond relaxation time approximation, the authors thoroughly investigated the current induced spin polarization in chiral tellurium and demonstrate the highly efficient charge-to-spin conversion. In particular, they proposed the concept of relaxons, which stand for the slow collective relaxation modes. The physical analysis appears thorough and systematic to me. I have the following major concerns:

- (1) As compared to their previous work Ref. 17 based on the constant relaxation time approximation, the theoretical method has been significantly improved. As such, the deeper analysis and understanding of charge-to-spin conversion are feasible. However, it seems to me that the similar study of current induced spin polarization in chiral systems using the Boltzmann transport equation beyond the relaxation time approximation has been performed in Ref. 23, which weakens the novelty of current work. The authors should clearly clarify the significant advances and novelty for this work as compared to previous work.
- (2) From Fig. 2e, the M_z values agree well with NMR experiment results. The authors should address how many parameters were used in their theoretical calculations.
- (3) From Fig. 2g, it seems to me that orbital angular momentum polarization dominates, which needs to be addressed.
- (4) In the introduction, the authors wrote that "in systems with equal Rashba and Dresselhaus parameters, spin accumulation cannot be induced by an electric current." This is not correct. The current induced spin polarization in PST has been demonstrated by previous work [see PRB 75, 155323 (2007); PRB 104, 085438 (2021)].
- (5) Since the point group at H point of chiral Te is D_3 , which sustains the pure Weyl Weyl-type SOC, I doubt the Hamiltonian Eq. 11 is symmetry adapted. Please specify.

(Remarks on code availability)

Reviewer #2

(Remarks to the Author)

The authors calculated the ratio of the spin polarization to the current density of Te using the first-principle calculation of electron structure and numerically obtained solution of the Boltzmann equation beyond relaxation time approximation. They found a quantitative agreement between their theoretical result and the experimental result of NMR measurement by Furukawa et al. The authors' achievement is significant in that Furukawa et al.'s experiment was the first observation of the current-induced spin polarization (so-called Edelstein effect) in the bulk of spatial inversion broken systems, and no previous studies quantitatively accounted for the experimental results. The combination of the first-principle calculation of electron structure and the exact solution of the Boltzmann equation endows the validity of the authors' scheme. The references are appropriately cited. The referee has some suggestions for presenting their results and arguments, which are listed below.

1. Figure 2e presents the main result of this manuscript. The reviewer suggests the authors present their results on spin polarization for a fixed current density together with the experimental results by Furukawa et al. and the theoretical results of the previous work by Roy et al. 2022 in the plot of a normal scale as well as the logarithmic scale. This revision clarifies how quantitatively agreement between the theory and experimental results is improved. In addition, in the normal plot, the

readers can compare the magnitudes M_z for $\mu = -20\text{meV}$ and -5meV more easily.

2. It would be beneficial to the readers if the authors explained how the theoretical value of the magnetization ratio to the current density is enhanced from those evaluated in the previous studies[17-19]. Figures 2a, b, and d indicate that the spin relaxation time is about $3\tau_{0}$, which would make the S_z triple that obtained by the conventional relaxation time approximation[17]. As the authors mentioned in p.7 in the manuscript, "Fig. 2g shows that the orbital contribution alone only doubles the total magnetization". Considering these two factors, the referee infers that M_z (present study) $\approx 6M_z$ (previous study). Is it correct?

3. At the end of the first paragraph on page 3, the authors raised the issue, "Can we find materials with strong SOC enabling both the efficient electrical generation of spin accumulation and long-range spin transport via a mechanism similar to PSH?" It is beneficial to the readers if the authors clearly state whether Te can support long-range spin transport based on quantitative reasoning or what they mean by "long-range transport." The spin diffusion length is roughly given by the product of spin relaxation time and the Fermi velocity. Thus, the referee speculates that the spin diffusion length is about triple the mean free path in Te. Does it imply long-range spin transport?

4. The sentence from Pages 12-13, "Remarkably, the pseudospin relaxation rate eventually drops to zero as Δ approaches small values. " may be reexamined. The vanishing relaxation rate of the pseudospin in the limit of $\Delta \rightarrow 0$ is a natural consequence of commutativity between τ_z and the Hamiltonian. Thus, the sentences quoted above should read as "Naturally, we confirm that the pseudospin relaxation rate eventually drops to zero as Δ approaches small values ".

(Remarks on code availability)

Reviewer #3

(Remarks to the Author)

Dear Editors,

Thank you for the opportunity to review the manuscript "Efficient Spin Accumulation Carried by Slow Relaxons in Chiral Tellurium" by Barts, E. et al. This study predicts an exceptional spin-to-charge conversion efficiency of 50% in chiral tellurium with a prolonged spin relaxation time, using the Boltzmann transport theory with the Relaxon formalism that is beyond the relaxation time approximation. The authors base their findings on the characteristics of tellurium's band structure near the valence band edge and Density Functional Theory calculations with tight-binding models derived from it to provide an accurate description of the electronic properties.

In my view, finding a material that reconciles the electrical generation of nonequilibrium spin densities with long-distance spin transport is crucial for advancing spintronics. Such a finding would undoubtedly merit publication in Nature Communications. However, I am not entirely convinced that the methodology employed by the authors effectively demonstrates this potential for chiral tellurium. Therefore, I cannot recommend the manuscript for publication unless the authors address the concerns outlined below.

Major Comments:

Boltzmann Transport Theory Application:

Major Comments:

Boltzmann Transport Theory Application:

While I have no major objections to using Boltzmann transport theory for computing nonequilibrium spin density, I have concerns regarding its application to orbital density and magnetization. Though appropriate for transition metal systems, this approximation does not yield correct results for p-orbital systems. The self-rotation of the wave function could play a significant role in chiral materials. The authors discuss how their methodology differs from traditional Boltzmann approaches but do not sufficiently address how the atom-centered approximation compares to modern theories. Moreover, the abstract mentions that the Relaxons dominate the spin and orbital angular momentum accumulation, but only the study focuses on the spin. I suggest looking into the orbital relaxation to complement the analysis. Additionally, it remains unclear whether the scalar disorder is diagonal on the orbital basis, which is critical for assessing the accuracy of their calculations.

Relaxation Time and Diffusive Coefficient:

The authors employ the Relaxon formalism to compute relaxation times. However, the correlation between these times and the diffusive coefficient or any experimental metric describing disorder is missing. This omission raises concerns about interpreting slow and fast Relaxons in this context. An estimate of the relaxation length would be beneficial to fully characterize the potential of chiral tellurium as a spintronics platform.

Comparison with Traditional Mechanisms:

The manuscript would benefit from a comparison between the proposed approach and traditional spin relaxation mechanisms such as D'yakonov-Perel' or Elliott-Yafet. This would provide the audience with a clearer understanding of the

novelty and implications of the results presented.

Robustness Against Fermi Energy Variations:

It is important to include a discussion on the robustness of the predicted quantities with respect to variations in the Fermi energy within the main text. This consideration would be highly relevant for experimentalists attempting to replicate or verify the findings.

(e) Minor Issues:

The acronym "SOF" is not defined in the manuscript.

On page 5, the parameters τ_0 , E_0 , and ρ_0 are introduced in a manner that implies they are mere parameters, whereas they should be presented as physical quantities with clear definitions.

In panel 3a, the color scheme and the dashed line make it difficult to follow the slow Relaxon relaxation time.

In panel 3d, there is an issue with the labels of the y and x axes, which are presented as the symbol for efficiency and τ_0 , respectively. Should not these be $S_z(t)$ and τ_0 ?

In the methods section, it is mentioned that the DFT energy states are projected onto the pseudo-atomic orbitals to build a tight-binding Hamiltonian. However, the orbitals used in this Hamiltonian are not specified nor are the basis of the effective model near the H point.

Conclusion:

In light of the concerns raised above, I regret that I cannot recommend this manuscript for publication in its current form. I encourage the authors to address these issues comprehensively to strengthen the manuscript and its contribution to the field of spin-orbitronics.

(Remarks on code availability)

In my opinion the code is reasonable to use and the instructions inside are clear enough.

Reviewer #4

(Remarks to the Author)

(Remarks on code availability)

Version 2:

Reviewer comments:

Reviewer #1

(Remarks to the Author)

I have carefully studied the authors' response letter and revised manuscript. While most of concerns are fully addressed, I still have the following comments before the manuscript may be considered for publication:

(1) Regrading comment (4), I would like to point that the current induced spin polarization in the PST (equal linear Rashba and Dresselhaus parameters) is indeed feasible. Evidently, the two Fermi contours with opposite spin textures have different Fermi wave vectors. As such, the current induced spin polarizations due to the displacements of two Fermi contours are not fully balanced. One may consider a simple case that the Fermi energy crosses only the lower branch of spin-split energy bands, the current induced nonequilibrium spin polarization is clearly finite. While I understand that this point is not very related to the subject of this work, I hope the authors would realize this.

(2) Regarding comment (5), the authors explain that the Hamiltonian (11) is defined in the pseudospin space, in while the basis is a complex linear combination of spin, orbital, and sublattice components. It is unclear to me how the authors can calculate the current induced spin $\langle S_z \rangle$ and orbital $\langle L_z \rangle$ density separately using Eq. (11). Please specify. It is also instructive to provide the calculation details for the expectation values of spin and orbit angular momentum operators using Eq. (11). In addition, Fig. 2b suggests that J_z is dominant while J_x and J_y are negligible, which is inconsistent with Fig. 2d that the arrows are not unidirectional along z direction. Those should be furthered elaborated.

(Remarks on code availability)

Reviewer #2

(Remarks to the Author)

The significance of the achievement of the present manuscript lies in the quantitative analysis of the magnetization of Chiral Telluride based on the first principle calculation of the electronic band structure (which was used in Ref. [17]) and the

spectral decomposition of the scattering integral (used in Ref. [23])

The author responds appropriately to my comments 1-4 in the previous report.

Particularly, the plot of the Magnetization curve in the normal scale that the authors added in the Supplemental material in the revised manuscript is more informative than the log-scale plot in the main text so that the readers can understand how the theoretical result in this manuscript is improved compared to the earlier result by Roy et al. [17] and how different the theoretical result in this manuscript is from the NMR result by Furukawa et al. [19].

An explicit remark on the spin diffusion length of the order of 100nm (line 276, line 353) is also informative so that the readers can understand what long-range spin transport means quantitatively.

In this way, the manuscript is improved. Before recommending the publication of this paper, however, the referee suggests three additional revisions, considering the significance of the results in the present manuscript, which can account quantitatively for the experimental results.

(1) The normal-scale plot is put in the main text, and the log-scale plot is put in the Supplemental material. The experimental result, the theoretical result in the present manuscript and the earlier theoretical results are within an order of magnitude.

(2) In comparison with the experimental result [19] for $jz=82\text{Am}^{-2}$ in Fig. 2e, theoretical results for $jz=82\text{Am}^{-2}$ instead of $jz=100\text{Am}^{-2}$ are more appropriate. As remarked by the authors in lines 151-154, the Rashba-Edelstein tensor has no free parameter, and thus it suffices to multiply the theoretical result by 0.82 without additional efforts of recalculation.

(3) Discussion on possible origins of discrepancy between the experimental and theoretical results in the present paper is better to be added. The theoretical results in the present paper have no free parameters based on the first-principle calculation and exact solution of the Boltzmann equation, and thus it seems at a glance nontrivial. It is beneficial to share the open issue with the readership.

(Remarks on code availability)

Reviewer #3

(Remarks to the Author)

I appreciate the authors' efforts in addressing my concerns and improving the manuscript. Their clarifications and additional discussions have strengthened their claims. As I mentioned previously, I believe the article is highly relevant, and while some concerns remain, the manuscript now presents a more compelling case for publication, which I therefore recommend. Below, I carefully evaluate whether the authors' revisions sufficiently address the issues raised in my initial review.

Boltzmann Transport Theory Application

The authors acknowledge the limitations of the atom-centered approximation in calculating the orbital contribution to the induced magnetization, which is particularly relevant for p-orbital systems like Te. While they claim to have implemented the modern theory of magnetization in PAOFLOW and obtained additional calculations, they have chosen not to incorporate these new findings in the revised manuscript, citing that a full treatment of orbital relaxation is beyond the scope of this work.

This justification is reasonable given that the manuscript primarily focuses on spin relaxation within the Relaxon formalism. The revised abstract and main text now make a clearer distinction between spin and orbital contributions, ensuring that the conclusions are not overstated. Additionally, the discussion acknowledges that orbital relaxation remains an open question, which was a necessary clarification.

Relaxation Time and Diffusive Coefficient

The revised manuscript now includes a whole page connecting their findings with experiments. While this addition improves experimental relevance, the direct connection between relaxon-derived relaxation times and measurable transport coefficients (such as mobility, resistivity, or diffusion constants) remains absent. The discussion mostly relies on order-of-magnitude estimates, rather than an explicit mapping between theory and experiment.

If feasible, a brief sentence acknowledging this limitation—e.g., "A direct mapping between relaxon lifetimes and experimental transport coefficients remains an open question for future work"—would strengthen the discussion without requiring additional calculations.

Comparison with Traditional Spin Relaxation Mechanisms

The revised manuscript now includes a comparison between the Relaxon-based spin relaxation and conventional spin relaxation mechanisms, such as the D'yakonov-Perel' and Elliott-Yafet mechanisms. This addition provides much-needed context, particularly for readers who may not be familiar with the Relaxon approach.

The authors now highlight that interband transitions, which are often important in conventional spin relaxation mechanisms, are suppressed in Te due to the large band splitting (~100 meV). This discussion is useful, as it clarifies why conventional coherent spin relaxation mechanisms do not play a dominant role in Te.

Although the comparison remains largely qualitative. A simple numerical estimate of how the spin relaxation time in Te compares to typical Dyakonov-Perel' and Elliott-Yafet timescales (even if only approximate) could make the discussion more concrete. Adding a short phrase, such as "For comparison, typical Dyakonov-Perel' relaxation times in III-V semiconductors range from X to Y ps, whereas our calculated spin relaxation time in Te is Z ps," would enhance clarity.

Robustness Against Fermi Energy Variations

The original concern was that the manuscript did not adequately discuss how charge-to-spin conversion efficiency and spin relaxation time vary with Fermi energy, which is an essential consideration for experimentalists. The revised version now explicitly addresses this point by clarifying the chemical potential dependence of the results.

The figures in the manuscript already included data for different chemical potential values before the revision, but the discussion has now been expanded. The new text explains that the chemical potential is the key experimental tuning parameter, particularly because Te crystals are naturally hole-doped with a typical chemical potential of ~ -20 meV. The added discussion ensures that readers understand the relevance of these values and their implications for real experimental setups.

Minor comments where fully addressed

Conclusion:

The revised manuscript successfully addresses most of the concerns raised in the initial review. The clarifications on orbital contributions, spin relaxation mechanisms, and Fermi energy dependence have strengthened the manuscript considerably. While some minor re-wordings would make the article more accurate, they are not important enough to prevent its publication. Therefore, I now find the manuscript suitable for publication.

(Remarks on code availability)

Reviewer #4

(Remarks to the Author)

(Remarks on code availability)

Groningen, Dec 10, 2024

Response letter for the manuscript NCOMMS-24-40416A-Z

Dear Editor,

We hereby resubmit our manuscript, *Efficient spin accumulation carried by slow relaxons in chiral tellurium*. We would like to thank the editor for handling our submission and selecting the referees with relevant expertise. We also thank the reviewers for their careful reading and constructive scientific feedback, which helped us to improve the manuscript significantly. In particular, we are glad that all referees recognized the importance of our work in condensed matter physics, materials science, and spintronics. We have analyzed all recommendations and criticisms and included the necessary clarifications in both the manuscript and the response below. The changes are highlighted in the resubmitted manuscript and discussed in the letter below. We are convinced that the revised version is suitable for publication.

Yours sincerely,

Evgenii Barts, Karma Tenzin and Jagoda Sławińska

Response to Reviewer 1

In this work, based on the advanced Boltzmann transport theory beyond relaxation time approximation, the authors thoroughly investigated the current-induced spin polarization in chiral tellurium and demonstrated the highly efficient charge-to-spin conversion. In particular, they proposed the concept of relaxons, which stand for the slow collective relaxation modes. The physical analysis appears thorough and systematic to me.

We appreciate that the reviewer finds our work thorough and systematic.

I have the following major concerns:

- 1. As compared to their previous work Ref. 17 based on the constant relaxation time approximation, the theoretical method has been significantly improved. As such, the deeper analysis and understanding of charge-to-spin conversion are feasible. However, it seems to me that a similar study of current-induced spin polarization in chiral systems using the Boltzmann transport equation beyond the relaxation time approximation has been performed in Ref. 23, which weakens the novelty of current work. The authors should clearly clarify the significant advances and novelty of this work as compared to previous work.*

We are glad that the reviewer recognizes the significant improvements in our theoretical approach compared to Ref. [17]. In that work, the authors used the constant relaxation time approximation to calculate current-induced spin accumulation and hypothesized the presence of

a persistent spin helix (PSH), which could potentially support long-range spin transport in chiral tellurium crystals. However, **two key questions remained unresolved**. First, the calculated induced magnetization was an order of magnitude smaller than the one measured experimentally using NMR, similar to recent theoretical studies showing the same discrepancy between theory and experiment. Second, without a direct calculation of spin relaxation time, the hypothesis of long-range spin transport remained unquantified. It was also unclear whether spin relaxation mechanisms mirror those observed in conventional systems, such as GaAs quantum wells.

In this manuscript, we achieve very good agreement with the experiments measuring the current-induced magnetization in Te. Through the exact solutions of the Boltzmann transport equations, which naturally account for the lifetime of excited collective electron states, we reveal that **the states with longer lifetimes carry spin accumulation, thereby enhancing the total spin accumulation lifetime and amplifying the charge-to-spin conversion efficiency**. Furthermore, we show how **these slow modes originate from the spin texture of the valence band and relate them to the persistent spin helix**. This new understanding of spin relaxation mechanisms is achieved by quantifying the transport of quasi-classical wavepackets with well-defined lifetime, termed relaxons, using the basis of wave functions derived from the accurate density functional theory calculations performed for Te. **It is the first time such an approach has been applied in spintronics**.

In contrast, Ref. [23] employs relaxons to describe spin relaxation within a two-dimensional electron gas model supplemented with anisotropic Weyl-type spin-orbit coupling. Although the authors made significant progress in advancing the concept of relaxons and successfully applied it to calculate spin accumulation, their results cannot be easily compared with the experiments. In particular, the model in Ref. [23] is not applicable to describe bulk Te, as it focuses on the spin accumulation at interfaces and it assumes the two-band electronic structure which is not the case in Te. In our method, we start with the electronic wave functions from density functional theory; therefore, **we do not rely on any model of the electronic structure**. Our approach, validated for Te, can be used for arbitrary materials, opening a way to identify realistic materials with long-range spin transport.

2. From Fig. 2e, the M_z values agree well with the NMR experiment results. The authors should address how many parameters were used in their theoretical calculations.

The calculated magnetization in our work relies on no free parameters, which makes our method more robust and suitable for predictions of quantities measured in experiments. We use the electron wavefunctions and energies from the accurate first-principles calculations, and the spin, orbital, and sublattice electron degrees of freedom are accounted for in the relaxation matrix within the collision integral of the Boltzmann equation. Our model does include a single free parameter, τ_0 , representing the characteristic timescale containing two introduced parameters that

describe the disorder potential: the density of impurities, and the strength of the scattering short-range potential that they induce. However, this free parameter cancels out in the Rashba-Edelstein tensor, as both the numerator and denominator in Eq.(9) are linearly dependent on τ_0 . To make this more evident, we introduced dimensionless time units by normalizing all time scales by τ_0 . Consequently, the calculated magnetization for a given value of charge current (e.g., $j_z = 100 \text{ Acm}^{-2}$) contains no free parameter, or, in other words, τ_0 remains implicitly incorporated into conductivity given by Ohm's law, $j_z = \sigma_{zz} E_z$. This procedure allows us to directly compare the calculated magnetization with the experimental value, which we explicitly mention in the revised manuscript.

The absence of free parameters is inherently due to the choice of the disorder potential. While we could have studied other types of potential, we believe this would add little value to the current understanding of the current-induced spin accumulation in Te. Also, this simplistic disorder potential is broadly accepted for modeling transport in clean crystalline samples. Most importantly, its simplicity helps to distill the main results, that is, the clear separation of the slow collective relaxation mode and its connection to the persistent spin helix.

It is worthwhile to note that τ_0 , a timescale unit in our model, remains a free parameter when calculating the spin relaxation time. It can be estimated from the mean free path. We elaborate on it in the response to reviewer 2 (point 3).

3. From Fig. 2g, it seems to me that orbital angular momentum polarization dominates, which needs to be addressed.

We thank the reviewer for this remark. In Fig. 2g, the plots show the induced spin and orbital density which do not include the corresponding g-factors equal to 2 and 1, respectively. The previous version of the caption of Fig. 2g mistakenly stated that it was the magnetization Mz . We have corrected the caption and we added information that the Sz and Lz contributions have to be multiplied by the g-factors to obtain the current-induced magnetization. The spin and orbital contributions to magnetization are therefore nearly equal, which is also stated in the text.

4. In the introduction, the authors wrote that "in systems with equal Rashba and Dresselhaus parameters, spin accumulation cannot be induced by an electric current." This is not correct. The current induced spin polarization in PST has been demonstrated by previous work [see PRB 75, 155323 (2007); PRB 104, 085438 (2021)].

We understand the reviewer's concern since this statement given in Ref. [16] (PRB 81, 085303 2010) was at first surprising for us. To clarify, we have rewritten this part of the introduction:

This is due to the symmetry restriction causing that in systems with exactly equal Rashba and Dresselhaus parameters, spin accumulation cannot be induced by an electric current, so spins still need to be injected using ferromagnetic electrodes or excited optically.

To further clarify, we note that the previous Boltzmann transport calculations mentioned by the reviewer [PRB 75, 155323 (2007); PRB 104, 085438 (2021)] demonstrated sizable current-induced spin accumulation in systems with equal Rashba (α_R) and Dresselhaus (α_D) parameters. However, the authors of Ref. [16] reproduced these results only in the vicinity of persistent spin texture (PST) (with slightly mismatched α_R and α_D) and reported zero effect precisely at PST ($\alpha_R = \alpha_D$). These findings were confirmed numerically via microscopically derived spin-and-charge diffusion equations and, most importantly, by symmetry-based arguments. The authors argued that at PST, the spin-orbit interaction, which can be treated as an effective gauge potential, becomes purely gauge and, therefore, can be gauged away through a unitary transformation of the electron wavefunctions, effectively making the spin degrees of freedom irrelevant.

Our interpretation of this effect from a semiclassical point of view is as follows. At PST, we have two twin copies of electron bands, which are shifted by the PSH wavevector in reciprocal space with respect to each other (see Ref. [7]). Within each band, the spin polarization is uniaxial and constant but with opposite signs between the two bands. This situation is analogous to having two parabolic free electron dispersions with opposite spin polarizations. In the semiclassical transport approach, spin mixing between these bands can only occur in the collision integral via interband scattering of wavepackets. This scattering, however, is prohibited because nonmagnetic impurities cannot flip spins, thus keeping the two bands decoupled. After observing this, we can consider a separate Boltzmann transport equation for electrons that belong to an individual band. Any induced shift in the electron distribution function, which leads to charge current, is represented by the populations of excited electrons and holes. However, the total induced spin polarization, the difference between the spin polarization of excited electrons and holes, is exactly zero because all states in this individual band have the same spin polarization.

5. Since the point group at H point of chiral Te is D_3 , which sustains the pure Weyl Weyl-type SOC, I doubt the Hamiltonian Eq. 11 is symmetry adapted. Please specify.

In Eq. (11) of this manuscript, we employ a $k \cdot p$ model to clarify the origin of the PSH in Te. This model has been broadly used in previous studies to describe the two valence bands and their gap opening [Ref. 18, 19, 50 and Averkiev *et al.*, Sov. Phys. Semicond. 18 (4), 402 (1984); Ivchenko and Pikus, Sov. Phys. Solid State 16 (7), 1261 (1974)]. The correct parameters that define this low-energy model, if they were unknown, could be extracted by fitting the model to the *ab initio* bands structure of the two valence bands. Although the reviewer is entirely correct that the point group of the H point (D_3) sustains the pure Weyl-type SOC, the $k \cdot p$ model

Hamiltonian from Eq. (11) remains symmetry-adapted because it acts in the pseudospin space, whose symmetry transformation rules are different as compared to spin. It is crucial to draw a clear distinction between spin and pseudospin degrees of freedom: in general, pseudospin is an effective degree of freedom that describes the spinor formed by the two valence band wavefunctions, which themselves are complex linear combinations of spin, orbital, and sublattice components. Therefore, the pseudospin operators do not have to transform in the same way as spin operators.

We have removed the confusing references to Weyl-type SOC from the last section.

Response to Reviewer 2

The authors calculated the ratio of the spin polarization to the current density of Te using the first-principle calculation of electron structure and numerically obtained solution of the Boltzmann equation beyond relaxation time approximation. They found a quantitative agreement between their theoretical result and the experimental result of NMR measurement by Furukawa et al. The authors' achievement is significant in that Furukawa et al.'s experiment was the first observation of the current-induced spin polarization (so-called Edelstein effect) in the bulk of spatial inversion broken systems, and no previous studies quantitatively accounted for the experimental results.

We thank the reviewer for recognizing our achievements.

The combination of the first-principle calculation of electron structure and the exact solution of the Boltzmann equation endows the validity of the authors' scheme. The references are appropriately cited. The referee has some suggestions for presenting their results and arguments, which are listed below.

- 1. Figure 2e presents the main result of this manuscript. The reviewer suggests the authors present their results on spin polarization for a fixed current density together with the experimental results by Furukawa et al. and the theoretical results of the previous work by Roy et al. 2022 in the plot of a normal scale as well as the logarithmic scale. This revision clarifies how quantitative agreement between the theory and experimental results is improved. In addition, in the normal plot, the readers can compare the magnitudes M_z for $\mu=-20\text{meV}$ and -5meV more easily.*

We have followed the suggestion of the reviewer. However, adding more content to Fig. 2e made it too crowded, and we created a separate figure, complementary to Fig.2e, in the Supplement.

- 2. It would be beneficial to the readers if the authors explained how the theoretical value of the magnetization ratio to the current density is enhanced from those evaluated in the*

previous studies [17-19]. Figures 2a, b, and d indicate that the spin relaxation time is about $3\tau_0$, which would make the S_z triple that obtained by the conventional relaxation time approximation [17]. As the authors mentioned in p.7 in the manuscript, “Fig. 2g shows that the orbital contribution alone only doubles the total magnetization”. Considering these two factors, the referee infers that M_z (present study) $\approx 6M_z$ (previous study). Is it correct?

The reviewer provides an elegant observation based on the qualitative analysis of the reported timescales. However, there is an additional detail that explains the reported order-of-magnitude enhancement compared to the previous studies. Our calculations show that the primary source of this enhancement comes from accurately taking into account the relaxation times of individual carriers. Specifically, relaxons that live longer contribute stronger to the distribution function and, consequently, to the magnetization. This relationship is reflected in Eq. (7) as a linear τ -dependence (after recalling that $1/\Gamma = \tau/\tau_0$). As the reviewer correctly pointed out, slow relaxons with lifetimes of about $3\tau_0$ carry spin and angular momentum, leading to the expected factor of 3 enhancement.

However, Eq. (7) also explicitly includes the relaxon eigenvector, derived from the projection of the initial electrically excited distribution function onto the relaxon basis. This introduces a correction to our estimation because the exact relaxon spectrum (see Fig.3c) reveals a more complex distribution of relaxons that define the initial non-equilibrium electron state.

In simpler terms, the two prominent peaks in Fig. 3c bear a resemblance to two collective relaxation modes, which can be thought of as wavepackets composed of individual relaxons. The mean lifetimes of these wavepackets are centered around 3τ and τ , which does boost the amplitude of the slower wavepacket. Nonetheless, their final contributions to the magnetization are governed by the exact amplitudes, shapes, and widths of these wavepackets, as provided by the exact mapping of the initial distribution function onto the relaxon basis.

Based on the suggestion of the reviewer, we have added the above discussion to the text.

- 3. At the end of the first paragraph on page 3, the authors raised the issue, “Can we find materials with strong SOC enabling both the efficient electrical generation of spin accumulation and long-range spin transport via a mechanism similar to PSH?” It is beneficial to the readers if the authors clearly state whether Te can support long-range spin transport based on quantitative reasoning or what they mean by “long-range transport.” The spin diffusion length is roughly given by the product of spin relaxation time and the Fermi velocity. Thus, the referee speculates that the spin diffusion length is about triple the mean free path in Te. Does it imply long-range spin transport?*

The reviewer reasonably assumes that the spin diffusion length is approximately given by the product of the spin relaxation time and the Fermi velocity. Therefore, the spin diffusion length in Te is estimated to be about three times the mean free path. Recent magnetoresistance experimental studies on Te nanoflakes [PRB 108, 115305 (2023)] have reported a mean free path of about 22-34 nm. Based on this, we estimate **the spin diffusion length to be 66-102 nm**. This value is comparable to the phase coherence length and spin-orbit relaxation length, which reach up to 100 nm, as obtained from weak localization model calculations in the cited reference. Such length scales are promising for crafting efficient spintronic devices that make use of non-local spin transport within two-dimensional heterostructures based on Te thin films or tellurene.

We have included these references in our revised manuscript. Since the question of whether Te can support long-range spin transport, in addition to effective charge-to-spin conversion, is important, we have addressed it in more detail in different sections of our revised manuscript.

- 4. The sentence from Pages 12-13, "Remarkably, the pseudospin relaxation rate eventually drops to zero as Δ approaches small values. " may be reexamined. The vanishing relaxation rate of the pseudospin in the limit of $\Delta \rightarrow 0$ is a natural consequence of commutativity between τz and the Hamiltonian. Thus, the sentences quoted above should read as "Naturally, we confirm that the pseudospin relaxation rate eventually drops to zero as Δ approaches small values ".*

We thank the reviewer for this remark. We have implemented it in the revised manuscript. We should add, however, that the limit of $\Delta \rightarrow 0$ recovers not only the U(1) rotational symmetry but also SU(2) full rotational symmetry, which is the essence of the persistent spin helix mechanism in GaAs quantum well [Ref. 18]. In this limit, the pseudospin Hamiltonian results in a unidirectional spin texture (e.g., in dimensionless units $H = p^2/2 - p_z \sigma_z$) that can be brought to a free electron gas Hamiltonian by a gauge transformation that restores the initially-hidden SU(2) symmetry (for $p' = p - \sigma_z$, $H = (p')^2/2$).

Response to Reviewer 3

Thank you for the opportunity to review the manuscript "Efficient Spin Accumulation Carried by Slow Relaxons in Chiral Tellurium" by Barts, E. et al. This study predicts an exceptional spin-to-charge conversion efficiency of 50% in chiral tellurium with a prolonged spin relaxation time, using the Boltzmann transport theory with the Relaxon formalism that is beyond the relaxation time approximation. The authors base their findings on the characteristics of tellurium's band structure near the valence band edge and Density Functional Theory calculations with tight-binding models derived from it to provide an accurate description of the electronic properties.

In my view, finding a material that reconciles the electrical generation of nonequilibrium spin densities with long-distance spin transport is crucial for advancing spintronics. Such a finding would undoubtedly merit publication in Nature Communications. However, I am not entirely convinced that the methodology employed by the authors effectively demonstrates this potential for chiral tellurium. Therefore, I cannot recommend the manuscript for publication unless the authors address the concerns outlined below.

We thank the reviewer for the careful reading of our work and the valuable feedback. We hope that the responses below and the revised manuscript allay the reviewer's doubts.

Major Comments:

- 1. Boltzmann Transport Theory Application: While I have no major objections to using Boltzmann transport theory for computing nonequilibrium spin density, I have concerns regarding its application to orbital density and magnetization. Though appropriate for transition metal systems, this approximation does not yield correct results for p-orbital systems. The self-rotation of the wave function could play a significant role in chiral materials. The authors discuss how their methodology differs from traditional Boltzmann approaches but do not sufficiently address how the atom-centered approximation compares to modern theories.*

We agree with the reviewer that our calculations of the orbital contribution to the induced magnetization are based on the atom-centered approximation which is less suitable for p-orbital systems. Inspired by the referee's remark, we have implemented the modern theory of magnetization in PAOFLOW and performed additional calculations which have indeed revealed a large influence of additional components on the orbital angular momentum accumulation. However, addressing these new findings, along with a comprehensive understanding of orbital relaxation, is beyond the scope of the manuscript revision and warrants a separate publication. It is important to note that the current work focuses on the relaxonic approach to the Boltzmann equation and the analysis of spin relaxation. While we included the orbital contribution using the simple approximation, and we keep it in the revised version, we now explicitly explain that this is only an initial step toward understanding orbital physics in Te. Spin and orbital relaxation are generally regarded as distinct processes, with the latter remaining veiled; there is considerable ongoing research in this area, which we intend to connect to our results in a future publication.

Moreover, the abstract mentions that the Relaxons dominate the spin and orbital angular momentum accumulation, but only the study focuses on the spin. I suggest looking into the orbital relaxation to complement the analysis.

As discussed above, the orbital accumulation is taken into account based on the atom-centered approximation. We indeed mostly focus on spin relaxation as stated in the abstract. In the revised

abstract, we attempted to make this distinction more explicit and emphasized that the relaxation was studied only for spin contribution to the total angular momentum accumulation.

Additionally, it remains unclear whether the scalar disorder is diagonal on the orbital basis, which is critical for assessing the accuracy of their calculations.

There is no need to make an ad hoc assumption that the scalar disorder is diagonal in the orbital basis. Instead, we directly calculate the scattering amplitudes using *ab initio* wavefunctions. The rationale behind our disorder potential choice is explained in the response to reviewer 1 (point 2).

2. Relaxation Time and Diffusive Coefficient: The authors employ the Relaxon formalism to compute relaxation times. However, the correlation between these times and the diffusive coefficient or any experimental metric describing disorder is missing. This omission raises concerns about interpreting slow and fast Relaxons in this context. An estimate of the relaxation length would be beneficial to fully characterize the potential of chiral tellurium as a spintronics platform.

The reviewer raised an important concern about interpreting and quantifying the predicted long-range transport in Te and its potential in spintronics. We believe that we fully addressed it in the response to reviewer 2 (point 3).

3. Comparison with Traditional Mechanisms: The manuscript would benefit from a comparison between the proposed approach and traditional spin relaxation mechanisms such as D'yakonov-Perel' or Elliott-Yafet. This would provide the audience with a clearer understanding of the novelty and implications of the results presented.

We thank the reviewer for this remark. In our revision, we added the discussion on coherent spin relaxation mechanisms, such as D'yakonov-Perel' and Elliott-Yafet, and compared spin relaxation in Te with these conventional mechanisms at the end of the section 'Spin accumulation lifetime'.

4. Robustness Against Fermi Energy Variations: It is important to include a discussion on the robustness of the predicted quantities with respect to variations in the Fermi energy within the main text. This consideration would be highly relevant for experimentalists attempting to replicate or verify the findings.

Our main results are presented with respect to different values of the chemical potential (the horizontal axis in Figs.1e, f, g, and the vertical axis in Fig.3 a, b), which is the essential metric that can be manipulated by electrostatic gating. Tellurium crystals are naturally doped with holes: a case that corresponds to the chemical potential of approximately -20 meV measured from the top of the valence band. At low temperatures, the chemical potential is effectively equivalent to

the Fermi energy. We have discussed in more detail the behavior of our predicted charge-to-spin conversion efficiency and spin relaxation time with respect to variations in the chemical potential and we provided the typical values of hole carrier density measured in the samples.

Minor Issues:

5. *The acronym "SOF" is not defined in the manuscript.*

We have fixed it in the revised version.

6. *On page 5, the parameters τ_0 , E_0 , and ρ_0 are introduced in a manner that implies they are mere parameters, whereas they should be presented as physical quantities with clear definitions.*

We have added explanations of the physical meaning of the parameters.

7. *In panel 3a, the color scheme and the dashed line make it difficult to follow the slow Relaxon relaxation time.*

We agree with the reviewer. We have modified the plot to improve the readability.

8. *In panel 3d, there is an issue with the labels of the y and x axes, which are presented as the symbol for efficiency and τ_0 , respectively. Should not these be $S_z(t)$ and τ_0 ?*

We have corrected it in the figure caption.

9. *In the methods section, it is mentioned that the DFT energy states are projected onto the pseudo-atomic orbitals to build a tight-binding Hamiltonian. However, the orbitals used in this Hamiltonian are not specified nor are the basis of the effective model near the H point.*

We followed the suggestion and added this information in the Methods section.

Conclusion:

In light of the concerns raised above, I regret that I cannot recommend this manuscript for publication in its current form. I encourage the authors to address these issues comprehensively to strengthen the manuscript and its contribution to the field of spin-orbitronics.

We trust that we have sufficiently addressed the concerns raised by the reviewer.

Groningen, Mar 9, 2025

Response letter for the manuscript NCOMMS-24-40416A-Z

Dear Editor,

We hereby resubmit our manuscript, *Efficient spin accumulation carried by slow relaxons in chiral tellurium*. We sincerely appreciate your handling our submission, and we thank the reviewers for their insightful feedback, which has helped us further strengthen the manuscript. We are encouraged that all reviewers acknowledge the significance of our work within a broad physical context, particularly in condensed matter physics, materials science, and spintronics. We have carefully addressed all additional remarks and suggestions, incorporating the necessary revisions, which are highlighted in the manuscript and detailed in our response below. We hope that the revised version meets the journal's standards and look forward to your consideration.

Yours sincerely,

Evgenii Barts, Karma Tenzin and Jagoda Sławińska

Response to Reviewer 1

I have carefully studied the authors' response letter and revised manuscript. While most of the concerns are fully addressed, I still have the following comments before the manuscript may be considered for publication:

We appreciate the valuable feedback from the Reviewer.

- 1. Regarding comment (4), I would like to point that the current induced spin polarization in the PST (equal linear Rashba and Dresselhaus parameters) is indeed feasible. Evidently, the two Fermi contours with opposite spin textures have different Fermi wave vectors. As such, the current induced spin polarizations due to the displacements of two Fermi contours are not fully balanced. One may consider a simple case that the Fermi energy crosses only the lower branch of spin-split energy bands, the current induced nonequilibrium spin polarization is clearly finite. While I understand that this point is not very related to the subject of this work, I hope the authors would realize this.*

We agree with the Reviewer that the issue of current-induced spin polarization in systems with equal linear Rashba and Dresselhaus parameters is subtle and less straightforward than we initially suggested. In our revised text, we primarily acknowledge and rely on previous works, aiming to introduce the known aspects of PST and broadly present the motivation for our study.

- 2. Regarding comment (5), the authors explain that the Hamiltonian (11) is defined in the pseudospin space, in while the basis is a complex linear combination of spin, orbital, and*

sublattice components. It is unclear to me how the authors can calculate the current induced spin $\langle Sz \rangle$ and orbital $\langle Lz \rangle$ density separately using Eq. (11). Please specify. It is also instructive to provide the calculation details for the expectation values of spin and orbit angular momentum operators using Eq. (11).

We do not calculate $\langle Sz \rangle$ and $\langle Lz \rangle$ using Eq. (11). The paper employs two different approaches, and the Boltzmann transport equations are solved for them independently. Our main results, including $\langle Sz \rangle$ and $\langle Lz \rangle$ (visualized in Fig. 2) and their lifetimes (Fig. 3), are obtained from the first approach, which is fully based on DFT wavefunctions and PAOFLOW calculations. We discuss this approach and the transport calculations in detail in the Introduction, the first section of the Results and the Methods.

Additionally, in the last section of the Results, we use the **kp**-model, which is defined in Eq. (11). This second model, being simpler, allows us to generalize the concept of slow relaxons to a broader class of materials. As shown in Fig. 4, we calculate only the induced pseudospin density $\langle \tau_z \rangle$ and its lifetime. Extracting the spin and orbital density from this model would be indeed difficult since it is an effective model formulated in pseudospin space.

As the Reviewer pointed out, using two approaches might be confusing for the reader. To clarify this, we modified the beginning of the **kp**-model section to draw a clear borderline between the two separate calculations. Also, we have modified Supplemental Note 1 to explain further how the spin and orbital expectation values are obtained using *ab initio* wavefunctions.

In addition, Fig. 2b suggests that J_z is dominant while J_x and J_y are negligible, which is inconsistent with Fig. 2d that the arrows are not unidirectional along z direction. Those should be further elaborated.

Fig. 2b shows the J_z polarization of the band dispersion exactly along the z axis in the vicinity of the H point ($k_x = 0, k_y = 0$), with $k_z > 0$ on the right of the H point and $k_z < 0$ on the left. This is consistent with Fig. 2d, where the arrows are unidirectional along exactly this line ($k_{[110]} = 0$ in terms of the crystallographic axes), which is, in fact, enforced by the crystal symmetry (see, e.g., S. B. Kilic *et al.* arXiv:2409.13632). Note that the arrows indicate only the direction of spin polarization and not its magnitude. We have clarified these points in the caption.

Response to Reviewer 2

The significance of the achievement of the present manuscript lies in the quantitative analysis of the magnetization of Chiral Telluride based on the first principle calculation of the electronic band structure (which was used in Ref. [17]) and the spectral decomposition of the scattering integral (used in Ref. [23])

The author responds appropriately to my comments 1-4 in the previous report.

Particularly, the plot of the Magnetization curve in the normal scale that the authors added in the Supplemental material in the revised manuscript is more informative than the log-scale plot in the main text so that the readers can understand how the theoretical result in this manuscript is improved compared to the earlier result by Roy et al. [17] and how different the theoretical result in this manuscript is from the NMR result by Furukawa et al. [19].

An explicit remark on the spin diffusion length of the order of 100 nm (line 276, line 353) is also informative so that the readers can understand what long-range spin transport means quantitatively.

In this way, the manuscript is improved. Before recommending the publication of this paper; however, the referee suggests three additional revisions, considering the significance of the results in the present manuscript, which can account quantitatively for the experimental results.

We appreciate the Reviewer' feedback on our revision and the new valuable comments that helped to refine our work.

- 1. The normal-scale plot is put in the main text, and the log-scale plot is put in the Supplemental material. The experimental result, the theoretical result in the present manuscript and the earlier theoretical results are within an order of magnitude.*

We have changed Fig.2 following the suggestion of the Reviewer.

- 2. In comparison with the experimental result [19] for $jz=82\text{Å}^{-2}$ in Fig. 2e, theoretical results for $jz=82\text{Å}^{-2}$ instead of $jz=100\text{Å}^{-2}$ are more appropriate. As remarked by the authors in lines 151-154, the Rashba-Edelstein tensor has no free parameter, and thus it suffices to multiply the theoretical result by 0.82 without additional efforts of recalculation.*

We have included this suggestion in the revised Fig.2 and in the section about magnetization.

- 3. Discussion on possible origins of discrepancy between the experimental and theoretical results in the present paper is better to be added. The theoretical results in the present paper have no free parameters based on the first-principle calculation and exact solution of the Boltzmann equation, and thus it seems at a glance nontrivial. It is beneficial to share the open issue with the readerships.*

We have added a detailed discussion in the third paragraph of the section about magnetization.

Response to Reviewer 3

I appreciate the authors' efforts in addressing my concerns and improving the manuscript. Their clarifications and additional discussions have strengthened their claims. As I mentioned previously, I believe the article is highly relevant, and while some concerns remain, the manuscript now presents a more compelling case for publication, which I therefore recommend. Below, I carefully evaluate whether the authors' revisions sufficiently address the issues raised in my initial review.

Boltzmann Transport Theory Application

The authors acknowledge the limitations of the atom-centered approximation in calculating the orbital contribution to the induced magnetization, which is particularly relevant for p-orbital systems like Te. While they claim to have implemented the modern theory of magnetization in PAOFLOW and obtained additional calculations, they have chosen not to incorporate these new findings in the revised manuscript, citing that a full treatment of orbital relaxation is beyond the scope of this work.

This justification is reasonable given that the manuscript primarily focuses on spin relaxation within the Relaxon formalism. The revised abstract and main text now make a clearer distinction between spin and orbital contributions, ensuring that the conclusions are not overstated. Additionally, the discussion acknowledges that orbital relaxation remains an open question, which was a necessary clarification.

Relaxation Time and Diffusive Coefficient

The revised manuscript now includes a whole page connecting their findings with experiments. While this addition improves experimental relevance, the direct connection between relaxon-derived relaxation times and measurable transport coefficients (such as mobility, resistivity, or diffusion constants) remains absent. The discussion mostly relies on order-of-magnitude estimates rather than an explicit mapping between theory and experiment.

If feasible, a brief sentence acknowledging this limitation—e.g., "A direct mapping between relaxon lifetimes and experimental transport coefficients remains an open question for future work"—would strengthen the discussion without requiring additional calculations.

We thank the Reviewer for this suggestion. We have incorporated it fully in the revised text at the end of the section "Spin accumulation lifetime and relaxon spectra".

Comparison with Traditional Spin Relaxation Mechanisms

The revised manuscript now includes a comparison between the Relaxon-based spin relaxation and conventional spin relaxation mechanisms, such as the D'yakonov-Perel' and Elliott-Yafet

mechanisms. This addition provides much-needed context, particularly for readers who may not be familiar with the Relaxon approach.

The authors now highlight that interband transitions, which are often important in conventional spin relaxation mechanisms, are suppressed in Te due to the large band splitting (~100 meV). This discussion is useful as it clarifies why conventional coherent spin relaxation mechanisms do not play a dominant role in Te.

Although the comparison remains largely qualitative. A simple numerical estimate of how the spin relaxation time in Te compares to typical Dyakonov-Perel' and Elliott-Yafet timescales (even if only approximate) could make the discussion more concrete. Adding a short phrase, such as "For comparison, typical Dyakonov-Perel' relaxation times in III-V semiconductors range from X to Y ps, whereas our calculated spin relaxation time in Te is Z ps," would enhance clarity.

We thank the Reviewer for this suggestion. We have included it in the revised manuscript.

Robustness Against Fermi Energy Variations

The original concern was that the manuscript did not adequately discuss how charge-to-spin conversion efficiency and spin relaxation time vary with Fermi energy, which is an essential consideration for experimentalists. The revised version now explicitly addresses this point by clarifying the chemical potential dependence of the results.

The figures in the manuscript already included data for different chemical potential values before the revision, but the discussion has now been expanded. The new text explains that the chemical potential is the key experimental tuning parameter, particularly because Te crystals are naturally hole-doped with a typical chemical potential of ~ -20 meV. The added discussion ensures that readers understand the relevance of these values and their implications for real experimental setups.

Minor comments were fully addressed

Conclusion:

The revised manuscript successfully addresses most of the concerns raised in the initial review. The clarifications on orbital contributions, spin relaxation mechanisms, and Fermi energy dependence have strengthened the manuscript considerably. While some minor re-wordings would make the article more accurate, they are not important enough to prevent its publication. Therefore, I now find the manuscript suitable for publication.

We are pleased that the reviewer acknowledges the improvements in our paper based on their valuable comments, and we appreciate their recommendation for its publication.